# Self-Efficacy Mediates the Effect of Framing Eating Disorders Prevention Message on Intentions to Have a Sufficient Weight: A Pilot Study

**DOI:** 10.3390/ijerph18178980

**Published:** 2021-08-26

**Authors:** Agnès Helme-Guizon, Marie-Laure Gavard-Perret, Rebecca Shankland, Valentin Flaudias

**Affiliations:** 1Laboratoire CERAG—Grenoble INP, Institut of Engineering and Management, Université Grenoble Alpes, 38000 Grenoble, France; marie-laure.gavard-perret@univ-grenoble-alpes.fr; 2Laboratoire DIPHE, Université Lumière Lyon 2, 69000 Lyon, France; rebecca.shankland@univ-lyon2.fr; 3Pôle Psychiatrie B, CHU Clermont-Ferrand, Université Clermont Auvergne, EA NPsy-Sydo, 63000 Clermont-Ferrand, France; vflaudias@chu-clermontferrand.fr

**Keywords:** self-efficacy, message framing, eating disorders, health prevention communication, dietary restraint intentions

## Abstract

Background: In the context of social marketing, the effectiveness of prevention messages is a major issue. The main objective of the present study was to assess the effect of prevention messages framing on self-efficacy reinforcement in order to improve intentions to reach or maintain sufficient weight in a non-clinical sample. It thus focuses on testing the mediating role of self-efficacy. Methods: Two hundred and thirty-three university student women were randomly assigned to one of the two conditions (gain-framed versus loss-framed message). They were exposed to a short persuasive message and surveyed on self-efficacy and intention to maintain sufficient weight. Results: Loss-framed messages elicited higher levels of self-efficacy than gain-framed messages, which led to higher intentions to reach or maintain sufficient weight. This study sheds light on the mediating role of self-efficacy. Conclusions: The results suggest ways to improve the persuasiveness of prevention campaigns, thereby opening up further research avenues.

## 1. Introduction

Eating disorders (ED) are becoming a public health concern and must be addressed through effective social marketing approaches, in particular because of their impact on individual physical and mental health, including high suicide rates [1], and their social and economic cost [2]. While nutritional attention may be positively associated with health, dieting and fasting have negative physical and mental consequences, lead to higher binge eating risks [3], and even lead to greater risk of being overweight [4]. To date, efficient ED prevention programs have been carried out in a group format [3,5,6,7,8]. However, their cost made them difficult to be generalized to the general population. Health promotion messages could be a means to broadly develop prevention in this field and, thus, a lever for action in social marketing, in particular through selective prevention campaigns aimed at a subgroup determined to be at high-risk [9].

In this perspective, perceived self-efficacy, defined as the perceived capability to perform a behavior [10], seems to play a determining role in persuading the individuals targeted by prevention messages. Often considered for its possible moderating role [11,12], self-efficacy was also shown to correlate with ED and seems to be a predictor of treatment outcome for eating disorder outpatients [13]. Moreover, as explained by Bandura, perceived self-efficacy is an important mediator in the process of behavior change, because it is influenced by various sources of information (vicarious experience, verbal persuasion, etc.), and in turn, it affects behaviors [13]. For example, in the context of children’s achievements, some authors underlined that observing models influenced their perceived self-efficacy and that, through this, their learning behaviors were improved [14]. In the same way, in the case of the relationship between attributions and exercise behavior, the perception of self-efficacy has been shown to be a mediator [15]. However, to our knowledge, only a few research studies initiated by Rimal specifically analyzed the mediating role of self-efficacy in the field of ED prevention [16,17]. Even if that research focused on a different relationship (between dietary knowledge and dietary behaviors) than the one we focus on in the present study, it highlights its potential mediator status. Its main conclusion was that prevention and health promotion campaigns should seek to directly address factors influencing self-efficacy. This is the main objective of the present study.

Past research showed that framing the same information either as gain (beneficial consequences) or as loss (detrimental consequences) leads to distinct behavioral decisions [18,19]. However, results concerning the most effective framing are inconsistent [20,21,22,23,24]. Kühberger [25] underlined that it is essential not only to understand “when” framing effect occurs but also “why” it does. Analyzing the mediating role of perceived self-efficacy between framing and intentions represents a means of better understanding the mechanisms of action of message framing. Although this has not been studied yet in this way, several articles based on different health-related contexts have shown a strong correlation between self-efficacy and intentions and/or behaviors, and some have shown the impact of self-efficacy beliefs on intentions and/or behaviors [26]. More specifically, concerning the mediating role of self-efficacy in the effect of framing on behaviors or intentions, negatively framed messages were shown to influence self-efficacy beliefs in the context of breast self-examination [27]. More generally, in other health issues such as human papillomavirus (HPV) vaccination messages, self-efficacy has been shown to mediate the relation between type of message and behavioral intentions [28]. In the field of healthy eating habits, self-efficacy has also been identified as a mediator of the intervention on changes in fruit and vegetable intake [29]. However, the contrasted results about the best framing and the lack of research on mediation by self-efficacy underline the need and importance of better understanding these mechanisms in order to promote appropriate eating disorder prevention messages. Moreover, the influence of gain vs. loss message framing on self-efficacy beliefs in the field of healthy eating habits has not yet been shown.

Regarding work on self-efficacy, a major recommendation by Bandura [10] was to design messages that make individuals aware of the seriousness of the problem and to thereby increase their sense of vulnerability. From this point of view, a negative framing is more likely to make people perceive this seriousness and vulnerability. In addition, the research led by Meyerowitz and Chaiken [27] clearly showed that “Only measures of perceived self-efficacy in performing BSE were differentially affected by the framing manipulation, with loss subjects reporting the greatest levels of self-confidence”. Researchers have also argued that persuasion may be enhanced through threatening messages (e.g., messages involving negative emotional appeals), as the message recipients will attempt to reduce the perceived threat and avoid potential negative consequences by acting in line with the recommendation. In adopting such a mindset, it may facilitate their belief in being able to adopt the suggested behavior [30]. Moreover, several studies suggest that negative information is considered more diagnostic and informative than positive information, particularly when the focus and processing are on message claims [31,32]. Thus, the perceived message could be treated as more reliable and accurate and would increase the sense of self-efficacy because the information relied upon would be perceived as less vague. In addition, in a study based on a thought-listing approach, participants expressed a need to improve their diet more often when exposed to a negatively framed (vs. positively framed) health message [33]. These authors justify this superiority of the negatively framed message due to its more persuasive nature and by providing another interesting explanation: a negatively framed message arouses greater “unrealistic optimism” in accordance with what some authors such as Gold and De Souza (2012) have shown [34]. These observations and explanations lead us to believe that negative framing, through its superior persuasion and, above all, through the triggering of unrealistic optimism, could more easily lead exposed individuals to develop a stronger belief in their ability to carry out the recommended behavior. Another interesting avenue for justifying a possible improvement in the perception of self-efficacy through negative framing could be that of the mental imagery elicited by the latter. Indeed, some authors such as Balbo and Gavard-Perret (2015) have clearly shown the difference in the valence of the mental imagery generated by each framing, with the negative framing being more likely to induce mental imagery of negative valence [35]. It is therefore possible that, due to the development of more negatively valenced mental imagery, individuals exposed to the negatively framed message are more likely to seek a reduction in the discomfort and unpleasantness associated with such imagery by more clearly asserting their self-efficacy to perform the behavior reducing the negative consequences [33]. In addition, this negatively valenced mental imagery may also be a lever for ‘unrealistic optimism’ in one’s self-efficacy beliefs [34]. In the same vein, one study indicated an interesting statistical trend regarding a positive (but not statistically significant) effect of mental imagery interventions on self-efficacy [36]. This assumption is also in line with the observation made by Rener et al. (2019): “Mental imagery allows us to ‘pre-experience’ future activities” [37]. This mental ‘pre-experience’ could then relate to the behavior to be implemented, which would thus appear easier to initiate and, as a result, would improve the self-efficacy belief. However, we can also think that, through the negative mental imagery generated by the loss framing, this negative framing would be more likely to induce a mental pre-experience of negative consequences. The mental discomfort felt would thus be increased and could encourage the individual to feel more motivated to believe in his or her abilities, in accordance with Meyerowitz and Chaiken (1987), in a logic of avoidance [27], even in the case where the mentally pre-experienced consequences are not life-threatening. Considering this, we postulated that loss-framed message (vs. gain-framed message) would have a more positive effect on intentions to maintain a sufficient weight through the enhancement of self-efficacy reported following message presentation.

## 2. Materials and Methods

### 2.1. Research Model

This study aimed to examine whether self-efficacy mediated the effect of loss- and gain-framed health messages on intention to reach or maintain a sufficient weight. The research model is shown in Figure 1.

### 2.2. Data Collection Procedure and Participants

This study is part of a broader study. Here, we focused on self-efficacy.

Participants: Prevalence data have identified female university students as a population at risk for ED for more than two decades [38,39]. Among adolescent girls, 75% report body dissatisfaction [40], and more than half of adolescent girls already report dietary restraints [41]. However, few research studies have been carried out on preventing ED through health communication messages for adolescents and for university students [42]. This population should become a privileged target since entering university corresponds not only to becoming more autonomous in terms of buying and cooking food but also to taking responsibility for oneself. Therefore, the targeted population was female, first-year university students. The inclusion criteria were being female, aged over 18, and enrolled in first year at university. No exclusion criteria were determined.

Procedure: Before conducting the study, we ran two pretests with the aim to ensure that the manipulation of gain- and loss-framed messages was effective, namely that the gain-framed message was perceived as more positive than the loss-framed message and vice versa.

Regarding the pilot study, female, first-year university students were recruited from February 22nd to March 4th 2011 on the campus of a French university via flyers and via emails sent to our students and to students from different disciplines thanks to emails relayed by faculties to their first-year students. The pilot study was conducted online using Survey Monkey software.

As a cover story, participants were informed that the study was about young women’s health. The experiment was carried out in accordance with The Code of Ethics of the World Medical Association (Declaration of Helsinki) for experiments involving humans. The participants were randomly assigned to one of the two conditions (gain-framed vs. loss-framed message). The participants were first asked questions related to the cover story about their physical activity, substance use, and dietary habits. They were then exposed to a message that consisted of a text, written in black and circled with green, a color that refers to health and hope in Europe [43,44]. In the gain-framed message condition (positive consequence), the message was: ‘Having a sufficient weight leads to better physical/intellectual performance’. In the loss-framed message condition (negative consequence), it stated: ‘Not having a sufficient weight reduces physical/intellectual performance’. The messages related to immediate gain or loss in order to foster higher involvement [45]. The message was deliberately vague regarding what the term ‘sufficient weight’ meant, as we were concerned about the fact that recommending a particular Body Mass Index (BMI) could evoke or induce restrictive behaviors. Finally, the participants were asked about their weight and size in order to calculate their BMI.

### 2.3. Measures

Self-efficacy was assessed using a single item adapted to the context of the study (as suggested by Bandura [10]): ‘You believe you are able to adjust your behaviors according to the message’. Intention to follow the message was also assessed through a single item: ‘You intend to reach or maintain a sufficient weight’. Both were measured on a 4-point Likert scale (1 = totally disagree to 4 = totally agree). Single item measurement was privileged since “time, response bias, participant fatigue, and ease of development [were] paramount” [46], and no common bias (inflated prediction) was evidenced [47].

### 2.4. Data Analysis

The data were analyzed using SPSS 25.0 (IBM, New York, NY, USA) and the PROCESS Macro program (Andrew F. Hayes, Alberta, Canada) as follows. First, message manipulations were checked by comparing the means of perceived valence of the message. Second, to test whether self-efficacy mediated the effects of message framing on intention to follow the recommendation (i.e., to have a sufficient weight), the PROCESS Macro [48] was selected, as it is more convenient than the series of separate analysis recommended by Baron and Kenny [49]. The PROCESS Macro method provides stronger and more robust results than with Baron and Kenny’s method and the Sobel test [50]. Third, the PROCESS Macro was also used to control for BMI effect.

These analyses were performed according to statistical recommendations [48,49]. The independent variable (message framing) is a binary variable using the following specification: 0 = ‘loss-framed’ and 1 = ‘gain-framed’. The mediation variable (self-efficacy) and the dependent variable (intention to follow the message) are continuous variables. In addition, the model included BMI as a control variable.

## 3. Results

### 3.1. Sample Characteristics

Two hundred and thirty-three questionnaires were collected. Twenty-six incomplete questionnaires were excluded. In total, 207 women fully completed the questionnaire, aged between 17 and 24 years old (M = 19, SD = 1.15), and with a BMI ranging from 16.56 to 33.26 (M = 21.44, SD = 3.09).

### 3.2. Manipulation Checks

In the gain-framed experimental condition, the message was perceived as being significantly more positive (Mpositive_consequence = 2.70, SD = 0.83, t (106) = 33.80, *p* < 0.001 and Mnegative_consequence = 1.84, SD = 0.81, t (106) = 23.38, *p* < 0.001). Conversely, in the loss-framed experimental condition, the message was perceived as being significantly more negative (Mpositive_consequence = 1.77, SD = 0.84, t (106) = 21.51, *p* <0.001; and Mnegative_consequence = 2.86, SD = 0.90, t (106) = 33.30, *p* < 0.001).

### 3.3. Self-Efficacy Mediation

To test the mediating role of self-efficacy on the effects of framing on intention to reach or maintain a sufficient weight, the SPSS Macro (PROCESS Model 4) was used as it was appropriate regarding our model (simple mediation). The results (Table 1), with a 5000-bootstrap sampling, revealed a significant effect of framing (loss vs. gain) on self-efficacy (*a* = 0.2576; *p* = 0.0126). Self-efficacy was higher (*F* = 6.334; *p* < 0.05) when the message was framed negatively (Mloss-framed = 3.08, SD = 0.764) than when framed positively (Mgain-framed = 2.82, SD = 0.762). Furthermore, the results showed that self-efficacy had a significant positive effect on intention to reach or maintain a sufficient weight (*b* = 0.4422; *p* = 0.0000). The more participants felt self-efficient, the more they expressed a high intention to reach or maintain a sufficient weight. Finally, framing (loss vs. gain) had no direct positive effect on intentions (*c’* = 1.0621; *p* = 0.2894). However, it had an indirect positive effect through the mediation of self-efficacy (*ab* = 0.4422 *p* = 0.000; IC [0.0271; 0.2254]) since the value in the 95% of confidence level did not include 0. Thus, self-efficacy fully mediated the relationship between message framing and behavioral intention, which supports the supposed mediating effect of self-efficacy.

### 3.4. Controlling for BMI

Furthermore, to rule out the possibility that BMI impacts self-efficacy, the SPSS Macro (PROCESS Model 7) was used as it was appropriate regarding our model (moderated mediation) (Figure 1). The interaction between message framing and BMI was not significant (a_31_ = 0.0034; *p* = 0.3981), indicating that BMI did not moderate the mediation by self-efficacy of the effects of message framing on intention to reach or maintain a sufficient weight.

## 4. Discussion

This study aimed to examine whether self-efficacy mediated the effect of loss- and gain-framed health messages on health behavioral intentions in the specific context of ED prevention. The focus on the mediating role of perceived self-efficacy, rarely examined in general and even less so in the specific context of ED, is an essential contribution in helping to understand the mechanisms at work in the effects of message framing. Our research provides an answer to the “why” of the latter as advocated by Kühberger [25]. More specifically, the results indicated that a loss-framed message was more persuasive, in terms of intentions to reach or maintain a sufficient weight, than a gain-framed message in the context of ED prevention aimed at reducing restrained eating in female, first-year university students. Furthermore, the results highlighted that a loss-framed prevention message fosters higher perception of self-efficacy and that the improvement of perceived self-efficacy is a way to obtain higher intentions to reach or maintain a sufficient weight. In summary, our results evidenced that self-efficacy mediates the effects of message framing on intention to follow the recommendation (i.e., to have a sufficient weight). This result underlines the potential usefulness of enhancing self-efficacy in health prevention campaigns, for example by showing how people similar to the targeted audience managed to change behavior (namely reduce dietary restrictions) and by providing models of efficacy behavior [51]. Individuals at risk of ED might therefore show more intention to follow loss-framed messages.

This pilot study did not intend to explore in depth or to identify the factors and explanatory mechanisms for the effect of framing on self-efficacy but rather only to validate the existence of this effect. However, we have mentioned several avenues to justify the interest of a negative framing in order to increase the perception of self-efficacy: in particular, diagnosticity and informativeness; aroused mental imagery; and unrealistic optimism. These are all avenues that should be explored in subsequent studies in order to better understand the mechanisms of action of loss-framing in terms of perceived self-efficacy.

This research is not without limitations. The tested messages did not offer a realistic health communication campaign context. Moreover, only one type of argument was tested: a positive (gain-framed) or negative (loss-framed) health outcome. Some authors advocate the use of social-based arguments, in particular in the context of youth prevention communication [52]. Further research should examine if the same effects hold with messages exploring other costs of dietary restraint behaviors (e.g., financial costs, deprivation, etc.). In addition, we used only one question to assess intention and self-efficacy. Future studies need more precise assessments of these dimensions. Furthermore, although our study was based on an experimental manipulation, the cross-sectional data collection does not allow for a direct test of the mediation effect, but it does provide information in this sense. Considering that the exploration of mediation in this context is uncommon, this effect will therefore have to be tested directly and experimentally in the future. Finally, we did not control whether participants were on a diet at the time of the survey or whether they had previously had dietary restraint behaviors. This could have influenced their perceived self-efficacy and their intentions to reach or maintain a sufficient weight.

## 5. Conclusions

Despite these limitations, the present pilot study highlighted the importance of exploring the mediating role of self-efficacy on intentions to follow recommendation messages (i.e., to maintain a sufficient weight), and the need to better understand how prevention messages may increase self-efficacy beliefs. This is particularly important considering the amount of obesity prevention messages that might encourage eating restriction and thus become a risk factor for ED. Moreover, the current pandemic and its repercussions in terms of ED are by now well-known and require special attention, especially among students [53,54]. Future studies need to better understand the psychological processes at play when using framed prevention messages in order to reconcile the prevention of obesity and the prevention of ED.

## Figures and Tables

**Figure 1 ijerph-18-08980-f001:**
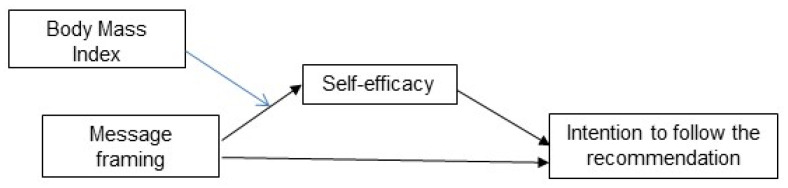
Research model.

**Table 1 ijerph-18-08980-t001:** PROCESS Macro mediation results.

Path	Indirect Effect	LLCI	ULCI
Framing → Self-Efficacy → Behavioral Intentions	0.4422 *	0.0271	0.2254

* *p* < 0.001.

## Data Availability

The data are subject to third party restrictions by the project funders.

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
