# Peer review of "Self-Efficacy Mediates the Effect of Framing Eating Disorders Prevention Message on Intentions to Have a Sufficient Weight: A Pilot Study"

_ijerph, 2021, doi:10.3390/ijerph18178980_

Round 1

Reviewer 1 Report

  1. The study is good. I suggest some modifications to the title.
  2. What is your sampling method?
  3. How did you come up with 233 participants?
  4. Do you have inclusion and exclusion criteria?
  5. What is the study period for each participant? Need to be mentioned.
  6. Your result indicated that a loss-framed message was more persuasive, in terms of intentions to reach or maintain sufficient weight than a gain-framed message in the context of ED prevention aiming at reducing restrained eating in the first-year university student women. Why? Could you justify it?
  7. I suggest re-writing of the conclusion part.

Reviewer 2 Report

Please see attached detailed comments.

Round 2

Reviewer 1 Report

Thank you so much for going through all the comments.

Author Response

We thank reviewer 1 for appreciating the improvements made to the manuscript.

Reviewer 2 Report

Please see attached summary.
